# On the Importance of Gradient Norm in PAC-Bayesian Bounds

**Itai Gat**[1], **Yossi Adi**[2,3], **Alexander Schwing**[4], **Tamir Hazan**[1]

[1] Technion
[2] FAIR Team, Meta AI Research
[3] The Hebrew University of Jerusalem
[4] University of Illinois at Urbana-Champaign

## Abstract

Generalization bounds which assess the difference between the true risk and the empirical risk have been studied extensively. However, to obtain bounds, current techniques use strict assumptions such as a uniformly bounded or a Lipschitz loss function. To avoid these assumptions, in this paper, we follow an alternative approach: we relax uniform bounds assumptions by using on-average bounded loss and on-average bounded gradient norm assumptions. Following this relaxation, we propose a new generalization bound that exploits the contractivity of the log-Sobolev inequalities. These inequalities add an additional loss-gradient norm term to the generalization bound, which is intuitively a surrogate of the model complexity. We apply the proposed bound on Bayesian deep nets and empirically analyze the effect of this new loss-gradient norm term on different neural architectures.

## 1   Introduction

Deep neural networks are ubiquitous across disciplines and often achieve state-of-the-art results. Despite the fact that deep nets are able to encode highly complex input-output relations, in practice, they do not tend to overfit [Zhang et al., 2016]. This tendency to not overfit has been investigated in numerous works on generalization bounds. Indeed, many generalization bounds apply to composite functions specified by deep nets. However, most of these results assume that the loss function satisfies various assumptions, such as uniformly bounded [Alquier et al., 2016], Lipschitz [Alquier et al., 2019], sub-Gaussian [Bégin et al., 2016, Alquier and Guedj, 2018], sub-gamma [Germain et al., 2016], or self-bounding [Haddouche et al., 2021]. Unfortunately, these assumptions exclude many state-of-the-art deep nets, whose properties cannot be statistically determined per data distribution.

In this work, we take a different route and use the deep net gradient-norm as a measure for PAC-Bayesian generalization. For this purpose, we treat the loss function as a random function that is generated by a high-dimensional probability space, which is governed by the generating process of the training data. This view allows us to introduce new tools from high-dimensional probability theory that relate measure concentration to the expansion of the loss function, as determined by its gradient-norm [van Handel, 2016].

We begin by adjusting a theorem (Herbst theorem) to estimate the measure concentration via the entropy of the loss function (Lemma 3.1). We then derive a term for the entropy tailored to the multiclass classification setting (discrete labels which condition Gaussian data), (Lemma 3.2). Finally we bound the entropy using a log-Sobolev inequality (Lemma 3.3).

Importantly, these steps result in a PAC-Bayesian bound which depends on the norm of the gradient of the loss. Intuitively this norm measures the complexity of the loss function, i.e., the model. Different from prior work, the bound hence depends on the structure of the employed model and its gradient

norm. This result admits a bound for multi-class classification with linear models and Lipschitz loss (Theorem 3.4), extending a result of Alquier et al. [2016], and multi-class classification with deep nets when the loss function and its gradient are *on-average* bounded (Theorem 3.5 and Theorem 3.6). The assumptions for the derived bound (Theorem 3.5 and Theorem 3.6) are closer to present-day deep net practice emphasizing the importance of the gradients in learning: in addition to the critical importance of controlling the loss gradient to better optimize a deep net, the gradient-norm also controls the generalization of the loss function (as measured by the deep net).

**Our contributions:** (1) We present a new PAC-Bayesian generalization bound that depends on the gradient-norm of the loss function. Consequently, we replace the uniform bound assumptions with an on-average bounded loss and an on-average bounded gradient norm. This answers an open problem raised by Bartlett et al. [2017a] (cf. Sec. 4) in the PAC-Bayesian setting. This result is presented in Theorem 3.5. (2) We extend the PAC-Bayesian bounds of Alquier et al. [2016] for the hinge-loss to any linear model with a Lipschitz loss function. This result appears in Theorem 3.4. (3) We empirically demonstrate that our bounds produce tighter generalization performance than the baseline methods. This helps to bridge the gap between theory and practice towards tighter bounds that have more realistic assumptions that match modern deep nets.

## 2 Background

Generalization bounds provide statistical guarantees on learning algorithms. They assess how the learned parameters $w$ of a model perform on test data given the model's result on the training data $S = \{(x_1, y_1), \ldots, (x_m, y_m)\}$, where $x_i$ is the data instance and $y_i$ is the corresponding label. The performance of the parametric model is measured by a loss function $\ell(w, x, y)$. The risk $L_D(w) = \mathbb{E}_{(x,y) \sim D} \ell(w, x, y)$ of this model is its average loss when the data instance and its label are sampled from the true but unknown distribution $D$. The empirical risk is the average training set loss $L_S(w) = \frac{1}{m} \sum_{i=1}^{m} \ell(w, x_i, y_i)$.

### 2.1 PAC-Bayesian bounds

PAC-Bayesian theory bounds the expected risk $\mathbb{E}_{w \sim q} L_D(w)$ of a model when its parameters are averaged over the learned posterior distribution $q$. The posterior distribution is learned from the training data $S$. In our work, we start from the following PAC-Bayesian bound:

**Theorem 2.1** (Alquier et al. [2016])**.** *Let $KL(q||p) = \int q(w) \log(q(w)/p(w)) dw$ be the KL-divergence between two probability density functions $p, q$. For any $\lambda > 0$, for any $\delta \in (0, 1)$ and for any prior distribution $p$, with probability at least $1 - \delta$ over the draw of the training set $S$, the following holds simultaneously for any posterior distribution $q$:*

$$\mathbb{E}_{w \sim q}[L_D(w)] \leq \mathbb{E}_{w \sim q}[L_S(w)] + \frac{1}{\lambda}[C(\lambda, p) + KL(q||p) + \log(1/\delta)], \tag{1}$$

*where $C(\lambda, p) \triangleq \log \left( \mathbb{E}_{w \sim p, S \sim D^m}[e^{\lambda(L_D(w) - L_S(w))}] \right)$.*

Unfortunately, the complexity term $C(\lambda, p)$ of this bound is challenging to compute exactly in general: it requires to integrate (and maximize if the log-sum-exp trick is used) over large parameter and data spaces. Using different assumptions, e.g., learning with sub-Gaussian or sub-gamma loss functions, learning with specific losses, etc., it is possible to analytically bound the complexity term $C(\lambda, p)$ as we briefly illustrate next.

We say that a loss function $\ell$ is sub-Gaussian with variance $\nu^2$ if it can be described by a sub-Gaussian random variable, i.e., if its log-moment generating function is upper bounded by $\nu^2/2$:

$$\log \left( \mathbb{E}_{(x,y) \sim D}[e^{\lambda(L_D(w) - \ell(w,x,y))}] \right) \leq \frac{\lambda^2 \nu^2}{2}. \tag{2}$$

Alquier et al. [2016] use Hoeffding's lemma to show that a uniformly bounded loss function, namely $0 \leq \ell(w, x, y) \leq B$ is a sub-Gaussian loss function with variance $B^2$ and hence derive from Theorem 2.1 and Equation (2) a PAC-Bayesian bound for bounded loss functions.

## 2.2 Measure concentration in high-dimension

However, the Hoeffding lemma, which asserts the sub-Gaussian property to a bounded loss function, treats the value of the loss as a real-valued random variable $\ell(w, x, y)$ while entirely ignoring the high-dimensional probability space $(x, y)$ that generates this random value. Said differently: any properties of a function $F$ transforming the data $(x, y)$ are ignored, and all functions $F$ are treated identically. This is sub-optimal. In the following, we describe an entropy method that utilizes the measure concentration phenomena that exist in the high-dimensional random space of $(x, y)$, cf. Chapter 3 of van Handel [2016]. The entropy of a non-negative random variable $F$ is

$$\text{Ent}[F] \triangleq \mathbb{E}[F \log F] - \mathbb{E}[F] \log \mathbb{E}[F]. \tag{3}$$

The Herbst theorem connects $\text{Ent}[F]$ to measure concentration by providing a bound on the log-moment generating function, as summarized next.

**Theorem 2.2** (Herbst). *Suppose that for all $\lambda > 0$ the following bound for the entropy holds:*

$$\text{Ent}[e^{\lambda F}] \leq \frac{\lambda^2 \sigma^2}{2} \mathbb{E}[e^{\lambda F}]. \tag{4}$$

*Then the scaled log-moment generating function which we also refer to as $\psi(\lambda)/\lambda$ is bounded as follows:*

$$\frac{\psi(\lambda)}{\lambda} \triangleq \frac{1}{\lambda} \log \left( \mathbb{E}[e^{\lambda F}] \right) \leq \mathbb{E}[F] + \frac{\lambda \sigma^2}{2}. \tag{5}$$

*Proof.* First, we note that due to the fundamental theorem of calculus

$$\frac{\psi(\lambda)}{\lambda} = \frac{\psi(0)}{0} + \int_0^\lambda \left( \frac{\psi(\alpha)}{\alpha} \right)' d\alpha. \tag{6}$$

Using l'Hopital rule, we verify that $\lim_{\alpha \to 0} \frac{\psi(\alpha)}{\alpha} = \mathbb{E}[F]$, which yields the first term of the bound. The following identity

$$\left( \frac{\psi(\alpha)}{\alpha} \right)' = \frac{\text{Ent}[e^{\alpha F}]}{\alpha^2 \mathbb{E}[e^{\alpha F}]}, \tag{7}$$

which is obtained from the definition of $\psi(\alpha)/\alpha$ and where $(\cdot)'$ refers to the derivative w.r.t. $\alpha$, derives the theorem after plugging Equation (7) into Equation (6) and using Equation (4) to bound the integral. □

Importantly, note that this is a step forward. Different from classical PAC-Bayesian bounds discussed in Section 2.1, which use the fact that a uniformly bounded loss function is sub-Gaussian, the Herbst theorem asserts that any random variable that satisfies Equation (4) is $\sigma^2$ sub-Gaussian.

To take advantage of this step, in this work, we benefit from the (modified) log-Sobolev inequality (LSI) for Gaussian random variables, which bounds the entropy.

**Theorem 2.3** (Gaussian log-Sobolev inequality (LSI), Gross [1975]). *Let $Z_1, ..., Z_d$ be independent Gaussian random variables, i.e., $Z_i \sim \mathcal{N}(\mu_i, \sigma_i^2)$. We then have*

$$\text{Ent}_{\mathcal{N}}[e^{f(Z_1, ..., Z_d)}] \leq \frac{1}{2} \mathbb{E}_{\mathcal{N}}[\|\sigma \cdot \nabla_Z f(Z_1, ..., Z_d)\|^2 e^{f(Z_1, ..., Z_d)}], \tag{8}$$

*where $\sigma \cdot \nabla_Z f(Z_1, ..., Z_d)$ is an element-wise multiplication.*

*Proof.* See supplementary material. □

One can apply the LSI to the Herbst theorem to attain measure concentration bounds for high-dimensional Lipschitz functions: a differentiable function $f(z_1, ..., z_d)$ is said to be $g$-Lipschitz if $|f(z_1, ..., z_d) - f(z_1', ..., z_d')| \leq g\|z - z'\|$, or equivalently if $\|\nabla f(z_1, ..., z_d)\| \leq g$ for any $z_1, ..., z_d$. The LSI for $g$-Lipschitz functions with $s \triangleq \sum_{i=1}^d \sigma_i^2$ variance reduces to the bound $\text{Ent}[e^{\lambda f(z_1, ..., z_d)}] \leq \frac{\lambda^2 g^2 s}{2} \mathbb{E}[e^{\lambda f(z_1, ..., z_d)}]$, which in turn provides a bound on the log-moment generating function of $f(z_1, ..., z_d)$ using Theorem 2.2.

# 3 PAC-Bayesian bounds with log-Sobolev inequalities

Different from prior work, we suggest measuring the generalization of neural nets by taking into account the expected gradient-norm of the loss function with respect to the data generation process.

For this, we note that classical PAC-Bayesian generalization bounds depend on measure concentration, as described in Theorem 2.1 and Equation (2.1). However, the sub-Gaussian assumption used in classical PAC-Bayesian generalization bounds seamlessly bypasses the measure concentration phenomena. This is too restrictive and enforces all loss functions, and consequently, all neural nets, to have the same measure concentration phenomena.

Instead, we *first* use the Herbst theorem to estimate the measure concentration by taking into account the entropy. In a *second* step we use Log-Sobolev inequalities (LSI) to bound the entropy, which allows us to consider the high-dimensional probability space $(x, y) \sim \mathcal{D}$ that controls the measure concentration of the loss function $\ell(w, x, y)$. This differs from the use of a crude worst-case bound on the loss function (e.g., $\ell(w, x, y) \leq B$ for any $w, x, y$) in classical PAC-Bayesian generalization bounds, which entirely ignores the neural net that generates the loss function.

**1) Herbst theorem to estimate measure concentration:** As just discussed, we first adjust the Herbst theorem to our setting:

**Lemma 3.1** (Herbst). *For any $\lambda > 0$ we have*

$$\log \left( \mathbb{E}_{S \sim \mathcal{D}^m} \left[ e^{\lambda(L_D(w) - L_S(w))} \right] \right) = \lambda \int_0^{\frac{\lambda}{m}} \frac{\mathrm{Ent}_{\mathcal{D}}[e^{-\alpha \ell(w,x,y)}]}{\alpha^2 \, \mathbb{E}_{\mathcal{D}}[e^{-\alpha \ell(w,x,y)}]} d\alpha. \tag{9}$$

*Proof.* First, we use the statistical independence of the training samples to decompose the moment generating function

$$M(\lambda) \triangleq E_{(x,y) \sim \mathcal{D}}[e^{\lambda(-\ell(w,x,y))}] \tag{10}$$

as follows:

$$\mathbb{E}_{S \sim \mathcal{D}^m} \left[ e^{\lambda(L_D(w) - L_S(w))} \right] = e^{\lambda L_D(w)} \mathbb{E}_{S \sim \mathcal{D}^m} [e^{\lambda \frac{1}{m} \sum_{i=1}^m (-\ell(w,x_i,y_i))}] \tag{11}$$

$$= e^{\lambda L_D(w)} \prod_{i=1}^m \mathbb{E}_{(x_i,y_i) \sim \mathcal{D}} [e^{\frac{\lambda}{m}(-\ell(w,x_i,y_i))}] \tag{12}$$

$$= e^{\lambda L_D(w)} M \left( \frac{\lambda}{m} \right)^m. \tag{13}$$

We apply the differential representation of Theorem 2.2 with $\psi(\lambda) = \log M(\lambda)$ and obtain:

$$\frac{\psi(\lambda)}{\lambda} = \int_0^\lambda \left( \frac{\psi(\alpha)}{\alpha} \right)' d\alpha + \frac{\psi(0)}{0} = \int_0^\lambda \frac{\mathrm{Ent}_{\mathcal{D}}[e^{-\alpha \ell(w,x,y)}]}{\alpha^2 \, \mathbb{E}_{\mathcal{D}}[e^{-\alpha \ell(w,x,y)}]} d\alpha - \mathbb{E}_{(x,y) \sim \mathcal{D}}[\ell(w, x, y)]. \tag{14}$$

Since $M(\lambda/m) = e^{\psi(\lambda/m)}$ and $\mathbb{E}_{(x,y) \sim \mathcal{D}}[\ell(w, x, y)] = L_D(w)$, we obtain from Equation (14) the identity

$$M \left( \frac{\lambda}{m} \right)^m = e^{\lambda \int_0^{\frac{\lambda}{m}} \frac{\mathrm{Ent}_{\mathcal{D}}[e^{-\alpha \ell(w,x,y)}]}{\alpha^2 \, \mathbb{E}_{\mathcal{D}}[e^{-\alpha \ell(w,x,y)}]} d\alpha - \lambda L_D(w)}, \tag{15}$$

which completes the proof when being combined with Equation (13). $\qquad\square$

**2) Entropy bound:** In the second step we now aim to bound the entropy $\mathrm{Ent}_{\mathcal{D}}[e^{-\alpha \ell(w,x,y)}]$.

Since we focus on the classification setting, the label $y$ is discrete. We further assume that for any (discrete) label $y$ its corresponding data $x = (x_1, ..., x_d)$ is generated from a $d$-dimensional Gaussian distribution that consists of $d$ i.i.d. Gaussian random variables $x_i \sim \mathcal{N}(\mu_{y,i}, \sigma_{y,i}^2)$. We denote this Gaussian distribution by $x \sim \mathcal{N}(\mu_y, \sigma_y^2)$ and abbreviate it by $x \sim \mathcal{N}_y$. The data generation distribution of $x$ is the Gaussian mixture model $\mathcal{D} = \sum_y \mathcal{D}_y \cdot \mathcal{N}_y$, where $\mathcal{D}_y$ is the marginal distributions of $\mathcal{D}$ w.r.t $y$.

We begin by decomposing the entropy of the (exponent) of the loss function according to the labels:

**Lemma 3.2.** *Let $\mathcal{D} = \sum_y D_y \cdot \mathcal{N}_y$ be a mixture model, whose label marginal probabilities are $D_y$ and set $f_w(x, y) = e^{-\alpha \ell(w,x,y)}$. One can show that*

$$\mathrm{Ent}_{\mathcal{D}}[f_w] = \sum_y D_y \, \mathrm{Ent}_{\mathcal{N}_y}[f_w] + \mathrm{Ent}_{\mathcal{D}_y}[\mathbb{E}_{x \sim \mathcal{N}_y}[f_w]]. \tag{16}$$

*Proof.* From the definition of the entropy of a function $\ell(w, x, y)$ with respect to its measure $(x, y) \sim \mathcal{D}$ we have

$$\mathrm{Ent}_{\mathcal{D}}[f] \triangleq \mathbb{E}_{\mathcal{D}} \, f_w(x, y) \log f_w(x, y) - (\mathbb{E}_{\mathcal{D}}[f_w(x, y)]) \log (\mathbb{E}_{\mathcal{D}}[f_w(x, y)]). \tag{17}$$

The result is attained by adding and subtracting the quantity $\mathbb{E}_{y \sim \mathcal{D}} \left( \mathbb{E}_{x \sim \mathcal{N}_y}[f_w(x, y)] \right) \log \left( \mathbb{E}_{x \sim \mathcal{N}_y}[f_w(x, y)] \right)$. The result then follows from the definitions of entropies of the mixed components, while setting $g(y) = \mathbb{E}_{x \sim \mathcal{N}_y}[f_w(x, y)]$. Specifically, we have

$$\mathrm{Ent}_{\mathcal{N}_y}[f_w] \triangleq \mathbb{E}_{\mathcal{N}_y}[f_w(x, y) \log f_w(x, y)] - \left( \mathbb{E}_{\mathcal{N}_y}[f_w(x, y)] \right) \log \left( \mathbb{E}_{\mathcal{N}_y}[f_w(x, y)] \right), \tag{18}$$

$$\mathrm{Ent}_{\mathcal{D}_y}[g] \triangleq \sum_y D_y g(y) \log(g(y)) - \left( \sum_y D_y g(y) \right) \log \left( \sum_y D_y g(y) \right). \tag{19}$$

$\square$

It is important to note that the number of components in the mixture model is not limited. Such a Gaussian mixture model can approximate any smooth density [Titterington et al., 1985, Scott, 1992, Goodfellow et al., 2016].

Next, we use the LSI to bound the loss function entropy using its expected gradient-norm with respect to the data generation process.

**Lemma 3.3.** *Assume that the loss per label is balanced, namely $\mathbb{E}_{x \sim \mathcal{N}_y}[e^{-\alpha \ell(w,x,y)}] = c$ for every $w$ and $y$, then under the conditions of Lemma 3.2.*

$$\log \left( \mathbb{E}_{S \sim \mathcal{D}^m} \left[ e^{\lambda(L_D(w) - L_S(w))} \right] \right) \le \frac{1}{2} \lambda \, \mathbb{E}_{\mathcal{D}} \left[ \|\sigma_y \cdot \nabla_x \ell(w, x, y)\|^2 \int_0^{\frac{\lambda}{m}} \frac{e^{-\alpha \ell(w,x,y)}}{\mathbb{E}_{\mathcal{D}}[e^{-\alpha \ell(w,x,y)}]} d\alpha \right]. \tag{20}$$

*Proof.* Lemma 3.2 asserts that $\mathrm{Ent}_{\mathcal{D}}[e^{-\alpha \ell(w,x,y)}]$ is composed of the entropies $\mathrm{Ent}_{\mathcal{N}_y}[e^{-\alpha \ell(w,x,y)}]$ and $\mathrm{Ent}_{\mathcal{D}_y}[\mathbb{E}_{x \sim \mathcal{N}_y}[e^{-\alpha \ell(w,x,y)}]]$. The assumption $\mathbb{E}_{x \sim \mathcal{N}_y}[e^{-\alpha \ell(w,x,y)}] = c$ implies the identity $\mathrm{Ent}_{\mathcal{D}_y}[\mathbb{E}_{x \sim \mathcal{N}_y}[e^{-\alpha \ell(w,x,y)}]] = \mathrm{Ent}_{\mathcal{D}_y}[c] = 0$. Thus $\mathrm{Ent}_{\mathcal{D}}[e^{-\alpha \ell(w,x,y)}] = \sum_y D_y \, \mathrm{Ent}_{\mathcal{N}_y}[e^{-\alpha \ell(w,x,y)}]$. We use Theorem 2.3 to bound

$$\mathrm{Ent}_{\mathcal{N}_y}[e^{-\alpha \ell(w,x,y)}] \le \frac{1}{2} \alpha^2 \, \mathbb{E}_{\mathcal{D}} \left[ \|\sigma_y \cdot \nabla_x \ell(w, x, y)\|^2 e^{-\alpha \ell(w,x,y)} \right]. \tag{21}$$

Combination with Lemma 3.1 concludes the proof. $\square$

Note, the per-label loss balance originates from bounding the term $\mathrm{Ent}_{\mathcal{D}_y}[\mathbb{E}_{x \sim \mathcal{N}_y}[f_w]]$ in Lemma 3.2. If the loss is per-label balanced, the entropy equals zero. In fact, the per-label balance can be relaxed by assuming that the loss is per-label bounded within an amplitude (Sec. 3.16 in van Handel [2016]).

Notably, the above lemma is more theoretical than practical: to estimate it in practice, one needs to avoid integrating over $\alpha$. Nevertheless, it serves as an essential generalization of the Herbst theorem and the LSI that is useful to derive PAC-Bayesian bounds for various settings. We look at these different settings next.

## 3.1 Linear models

In the following, we consider differentiable and Lipschitz loss functions[1] over linear models in a multi-class setting of the form $\ell(w, x, y) \triangleq \hat{\ell}(Wx, y)$. Included in these assumptions are the popular NLL loss $-\log p(y|x, w) = -(Wx)_y + \log(\sum_{\hat{y}} e^{(Wx)_{\hat{y}}})$ that is used in logistic regression and the multi-class hinge loss $\max_{\hat{y}}\{(Wx)_{\hat{y}} - (Wx)_y + \mathbb{1}[y \ne \hat{y}]\}$ that is used in support vector machines.

---

[1] It is enough to assume that the loss function is continuous and almost everywhere differentiable with respect to $x$.

**Theorem 3.4.** *Let $\ell(w, x, y) \triangleq \hat{\ell}(Wx, y)$ be a differentiable loss function over $x = (x_1, ..., x_d)$ and $k$ classes $y \in \{1, ..., k\}$, with Lipschitz constant $g$, i.e., $\|\nabla_t \hat{\ell}(t, y)\| \leq g$. Consider a Gaussian prior distribution $p \sim N(0, \sigma_p^2)$. Under the conditions of Lemma 3.3, for any $0 < \lambda \leq \sqrt{\frac{m}{16}}/g\sigma_p\sigma_y$ and $\delta \in (0, 1)$, with probability at least $1 - \delta$ over the draw of the training set $S$, we have*

$$\mathbb{E}_{w \sim q}[L_D(w)] \leq \mathbb{E}_{w \sim q}[L_S(w)] + \frac{kd \log(\sqrt{4/3}) + KL(q\|p) + \log(1/\delta)}{\lambda}. \tag{22}$$

*Proof.* This bound is derived by applying Lemma 3.3. We begin by realizing the gradient of $\hat{\ell}(Wx, y)$ with respect to $x$. Using the chain rule, $\nabla_x \hat{\ell}(Wx, y) = W^\top \nabla_{Wx} \hat{\ell}(Wx, y)$. Hence, we obtain for the gradient norm $\|\nabla_x \hat{\ell}(Wx, y)\|^2 \leq \|\nabla_{Wx} \hat{\ell}(Wx, y)\|^2 \cdot \sum_{y=1}^{k} \sum_{j=1}^{d} w_{y,j}^2 \leq g^2 \sum_{y=1}^{k} \sum_{j=1}^{d} w_{y,j}^2$. Plugging this result into Lemma 3.3, we obtain the following bound:

$$\mathbb{E}_{\mathcal{D}} \left[ \|\sigma_y \nabla \ell(w, x, y)\|^2 \int_0^{\frac{\lambda}{m}} \frac{e^{-\alpha\ell(w,x,y)}}{M(\alpha)} d\alpha \right] \leq \sigma_y^2 g^2 \sum_{y=1}^{k} \sum_{j=1}^{d} w_{y,j}^2 \int_0^{\frac{\lambda}{m}} \frac{\mathbb{E}_{\mathcal{D}}\left[ e^{-\alpha\ell(w,x,y)} \right]}{M(\alpha)} d\alpha.$$

Since $M(\alpha) \triangleq \mathbb{E}_{\mathcal{D}}\left[ e^{-\alpha\ell(w,x,y)} \right]$, the ratio in the integral equals one and the integral $\int_0^{\frac{\lambda}{m}} d\alpha = \frac{\lambda}{m}$. Finally, whenever $\lambda g\sigma_p\sigma_y \leq \sqrt{m/4}$ we follow the Gaussian integral and derive the bound

$$\log \left( \mathbb{E}_{w \sim p} e^{\frac{\sigma_y^2 \lambda^2 g^2}{2m} \sum_{y,j} w_{y,j}^2} \right) \leq kd \cdot \log \left( \sqrt{\frac{m}{m - m/4}} \right) = kd \cdot \log(\sqrt{4/3}). \tag{23}$$

Finally, we obtain Equation (22) by plugging the above into Theorem 2.1. A detailed proof is provided in Appendix. □

Notice, in the linear case, the gradients of the model are a function of $W$. Thus they are not uniformly bounded for every $W$. Since we assume the loss to be Lipschitz, we can bound $C(\lambda, p)$, and by further assuming bounded $\lambda$, and Gaussian $W$, we obtain $kd \cdot \log(4/3)$.

Theorem 3.4 provides a PAC-Bayesian bound for classification using the NLL loss. This extends the result of Alquier et al. [2016] for binary hinge-loss to the multi-class hinge loss (cf. [Alquier et al., 2016], Sec. 6).

While the above result can be applied to deep nets, obtaining a value for the bound requires computing the Lipschitz constant, which is an NP-hard problem even for two-layer neural nets [Virmaux and Scaman, 2018]. Moreover, common Lipschitz constant approximation algorithms tend to overestimate the constant and are exponential in the network's depth [Virmaux and Scaman, 2018, Combettes and Pesquet, 2019].

## 3.2 Non-linear models

Sadly, the bound proposed in the previous sub-section cannot be applied to non-linear loss functions since their gradient-norm is not bounded, as evident by the exploding gradients property in deep nets. To mitigate this, we suggest utilizing on-average bounded losses and gradients. Formally, one can show the following:

**Theorem 3.5.** *Consider smooth loss functions that are on-average bounded, i.e., for every $w$ the following holds: $\mathbb{E}_{\mathcal{D}} \ell(w, x, y) \leq b$ and $\mathbb{E}_{\mathcal{D}} \left[ \|\nabla_x \ell(w, x, y)\|^2 \right] \leq g$. Under the conditions of Lemma 3.3 for any $0 < \lambda \leq m$ and $\delta \in (0, 1)$, with probability at least $1 - \delta$ over the draw of the training set $S$, we obtain*

$$\mathbb{E}_{w \sim q}[L_D(w)] \leq \mathbb{E}_{w \sim q}[L_S(w)] + \frac{\frac{\lambda^2 e^b g \sigma_y^2}{2m} + KL(q\|p) + \log(1/\delta)]}{\lambda}. \tag{24}$$

*Proof.* This bound is derived by applying Lemma 3.3 and bounding $\int_0^1 \frac{e^{-\alpha\ell(w,x,y)}}{M(\alpha)} d\alpha \leq e^b$. We derive this bound in three steps: First, from $\ell(w, x, y) \geq 0$ we obtain $e^{-\alpha\ell(x,x,y)} \leq 1$. Then, we lower bound $M(\alpha) \geq M(1)$ for any $0 \leq \alpha \leq \lambda/m$. Also, since $\ell(w, x, y) \geq 0$ the function $e^{-\alpha\ell(w,x,y)}$ is monotone in $\alpha$ within the unit interval and $M(\alpha) \geq M(1)$ for any $\alpha \leq 1$. Lastly, the assumption

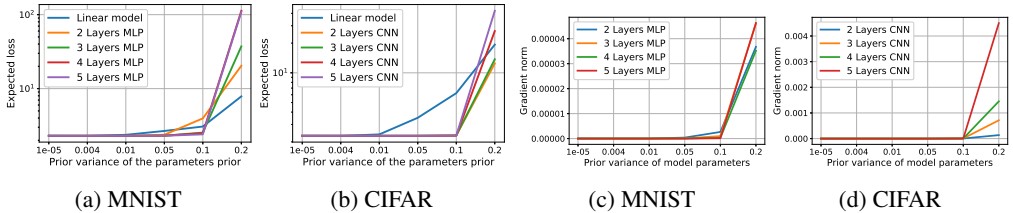

(a) MNIST      (b) CIFAR      (c) MNIST      (d) CIFAR

Figure 1: (a-b) $\mathbb{E}_{\mathcal{D}}[\ell(w, x, y)]$ as a function of different priors for the model parameters. (c-d) $\mathbb{E}_{\mathcal{D}}\left[\|\nabla_x \ell(w, x, y)\|^2\right]$ as a function of the different variance levels of the prior distribution. Results are reported for MNIST (a, c) using MLPs, and CIFAR10 (b, d) using CNNs.

$\mathbb{E}_{\mathcal{D}}[-\ell(w, x, y)] \geq -b$ and the monotonicity of the exponential function result in the lower bound $e^{\mathbb{E}_{\mathcal{D}}[-\ell(w,x,y)]} \geq e^{-b}$.

From convexity of the exponential function we have $\mathbb{E}_{\mathcal{D}} e^{-\ell(w,x,y)} \geq e^{\mathbb{E}_{\mathcal{D}}[-\ell(w,x,y)]}$. Combining these bounds and replacing $\mathbb{E}_{\mathcal{D}}\left[\|\sigma_y \nabla \ell(w, x, y)\|^2\right]$ with $g\sigma_y^2$, we obtain the upper bound

$$C(\lambda, p) \leq \frac{\lambda^2 e^b g \sigma_y^2}{2m}. \tag{25}$$

Finally, we obtain Equation (24) by plugging Equation (25) into Theorem 2.1. A detailed proof is provided in Appendix. □

In contrast to prior work that assumes the loss function to be uniformly bounded, i.e., $\ell(w, x, y) \leq B$ for every $w, x, y$ [Alquier et al., 2016], in Theorem 3.5, we assume an on average bounded loss and an on average bounded gradient norm, i.e., $\mathbb{E}_{\mathcal{D}} \ell(w, x, y) \leq b$ and $\mathbb{E}_{\mathcal{D}}\left[\|\nabla \ell(w, x, y)\|^2\right] \leq g$ for all $w$.

The above bound corresponds to an open problem raised by Bartlett et al. [2017a], wondering about the existence of generalization bounds that assume on-average bounds on the loss and/or the gradient norm. Nevertheless, this bound is more theoretical than practical, as it enforces global on-average bounds $b \geq \mathbb{E}_{\mathcal{D}} \ell(w, x, y)$ and $g \geq \mathbb{E}_{\mathcal{D}}\left[\|\nabla \ell(w, x, y)\|^2\right]$.

For a more practical perspective, in the following, we consider the on-average loss and gradient norm to be functions of the model parameters.

**Theorem 3.6.** *Consider smooth loss functions and the conditions of Lemma 3.3. For any $0 < \lambda \leq m$ we obtain*

$$C(\lambda, p) \leq \log\left(\mathbb{E}_{w \sim p}\left[\exp\left(\frac{\lambda^2 e^{\mathbb{E}_{\mathcal{D}} \ell(w,x,y)} \mathbb{E}_{\mathcal{D}}\left[\|\nabla \ell(w, x, y)\|^2\right] \sigma_y^2}{2m}\right)\right]\right). \tag{26}$$

The proof of this Theorem follows the proof of Theorem 3.5. However, in this case, we cannot remove the expectation over the prior distribution since we assume the bound over the loss and gradients to depend on $w$.

The above derivation upper bounds the complexity term $C(\lambda, p)$ by the expected gradient-norm of the loss function, i.e., the flow of its gradients through the model's architecture. We show empirically that the rate of the bound $\lambda$ can be as high as $m$, dependent on the gradient-norm. This is a favorable property since the convergence of the bound scales as $1/\lambda$. Therefore, one would like to avoid exploding gradient-norms, which effectively harm the true risk bound. While one may achieve a fast rate bound by forcing the gradient-norm to vanish rapidly, practical experience shows that vanishing gradients prevent the deep net from fitting the model to the training data when minimizing the empirical risk. In our experimental evaluation, we demonstrate the influence of the expected gradient-norm on the bound of the true risk.

## 4 Experiments

In this section, we evaluate our PAC-Bayesian bounds experimentally, both for linear and non-linear models. We begin by verifying our assumptions, comparing the proposed bound to prior work, and

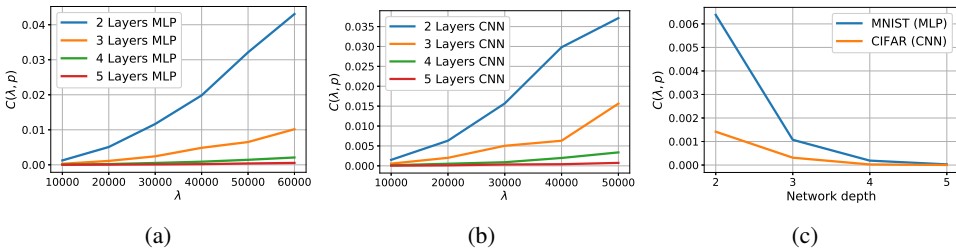

(a)           (b)           (c)

Figure 2: Complexity study: In (a-b), we present the complexity term as a function of $\lambda$. Furthermore, in (c), we present the complexity term $C(\lambda, p)$ as a function of the network depth for both CIFAR and MNIST. Note, the bound on the complexity term decreases as a function of the depth of the network.

estimating its predictive generalization capabilities. Next, we study the behavior of the complexity term $C(\lambda, p)$ for different architectures, both for linear models and deep nets. We conclude the section with an evaluation of the effectiveness of the proposed bound at predicting generalization performance and analyzing its different components during optimization. All reported results were averaged over three runs using different seeds. The complete experimental setup can be found in the Appendix.

**Verifying assumptions:** In Lemma 3.3 we assume that the loss per label is balanced. To verify that this assumption holds, we use ten different architectures (ResNet18, PreActResnet18, GoogLeNet, VGG11, VGG13, VGG16, VGG19, DenseNet121, MobileNet, EfficientNetB0) on CIFAR10 and CIFAR100 [Krizhevsky, 2009, Simonyan and Zisserman, 2014, Szegedy et al., 2015, He et al., 2016, Huang et al., 2017, Howard et al., 2017, Tan and Le, 2019]. The maximum standard deviation across the labels is 0.022, while the mean value is 4.605. Hence, it is evident that this assumption holds in practice. In Theorem 3.6 we assume that the loss is unbounded but it is on-average bounded by a function depending on $w$, i.e., $\mathbb{E}_{\mathcal{D}} \ell(w, x, y) \leq b(w)$. The results to verify this for MNIST [LeCun et al., 1998] and CIFAR10 are provided in Fig. 1a and Fig. 1b. We observed that until $\sigma_p^2 = 0.1$, the loss is on-average bounded by $\sim 2$. Moreover, for $\sigma_p^2 \leq 0.01$, the on-average loss bound is about 1, and its effect on the complexity term $C(\lambda, p)$ is minor. This validates empirically our assumptions that the on-average bounds $\mathbb{E}_{\mathcal{D}} \ell(w, x, y)$ are small although $\max_{w,x,y} \ell(w, x, y)$ is much larger (see Tab. 1 for its impact on the generalization).

**Complexity of neural nets:** We now turn to estimate our bounds over $C(\lambda, p)$, both for linear and non-linear models. Recall that $\lambda$ determines the rate of the generalization bound, and one would like to set $0 < \lambda \leq m$ as large as possible while $C(p, \lambda)$ to be as small as possible. As the bound on $C(\lambda, p)$ is controlled by the expected gradient-norm $\mathbb{E}_{\mathcal{D}} \left[\|\nabla_x \ell(w, x, y)\|^2\right]$, we present in Fig. 1c and Fig. 1d the expected squared gradient-norm as a function of different variance levels for the prior distribution $p$ over the model parameters, again using both MNIST and CIFAR10 data.

We note that the linear model has the largest expected gradient-norm. Further, note that the deeper the network, the smaller its gradient-norm. As a result, deeper nets can use larger values of $\lambda$ for the generalization bound. Thus, we present our complexity bound as a function of $\lambda$ in Fig. 2a and Fig. 2b. Note, the complexity term $C(\lambda, p)$ reaches its minimum around $\lambda = m$. Fig. 2c studies the effect of a network's depth on the complexity term. We observe that, as expected based on the earlier plots, deeper networks have a lower complexity term. We attribute this phenomenon to vanishing gradients which create a contractivity property that stabilizes the loss function, i.e., reduces its variability. However, this comes at the expense of the expressivity of the deep net, since deep nets with vanishing gradients cannot fit the training data in the learning phase.

**Comparison to prior PAC-Bayesian bounds:** We compare the proposed bound to Alquier et al. [2016] (bounded version), Germain et al. [2016], and Dziugaite and Roy [2017] on both MNIST and CIFAR. Unlike our bound which assumes the expected loss to be bounded, both Alquier et al. [2016] and Germain et al. [2016] assume the loss to be uniformly bounded by a constant $B$. Hence, for a fair comparison, we estimated $B$ to be the maximum and average loss over the training set and use it to bound the maximum and expected loss respectively. Results are summarized in Tab. 1. Notice, our bound produces tighter generalization bounds compared to the baseline methods for both datasets.

Table 1: Comparison of the full PAC-Bayesian generalization bound versus Alquier et al. [2016], Germain et al. [2016], and Dziugaite and Roy [2017]. For MNIST we use a three layer fully connected model, while for CIFAR-10 we use a CNN with three convolutional layers. In all settings we used $\lambda = m$ and $\sigma_p^2 = 0.01$.

| DATASET | ALQUIER | GERMAIN | DZIUGAITE | OURS |
|---------|---------|---------|-----------|------|
| MNIST | $3.36 \pm$ 5E-4 | $1.86 \pm$ 4E-3 | $0.41 \pm$ 3E-4 | **$0.04 \pm$ 5E-4** |
| CIFAR-10 | $4.81 \pm$ 1E-2 | $8.37 \pm$ 9E-2 | - | **$0.05 \pm$ 4E-3** |

## 5   Related work

Generalization bounds for deep nets were explored in various settings. VC-theory provides both upper bounds and lower bounds to the network's VC-dimension, which are linear in the number of network parameters [Bartlett et al., 2017b, 2019]. While VC theory asserts that such a model should overfit as it can learn any random labeling (e.g., [Zhang et al., 2016]), surprisingly, deep nets generally do not overfit.

Rademacher complexity allows to apply data-dependent bounds to deep nets [Bartlett and Mendelson, 2002, Neyshabur et al., 2015, Bartlett et al., 2017a, Golowich et al., 2017, Neyshabur et al., 2018]. These bounds rely on the loss and the Lipschitz constant of the network and consequently depend on a product of norms of weight matrices, which scales exponentially in the network depth. Wei and Ma [2019] developed a bound over the gradient-norm of training examples. In contrast, our bound depends on average quantities of the gradient-norm. Thus we answer an open question of Bartlett et al. [2017a] about the existence of bounds that depend on average loss and average gradient-norm, albeit in a PAC-Bayesian setting. PAC-Bayesian bounds that use Rademacher complexity have also been studied [Kakade et al., 2009, Yang et al., 2019].

Stability bounds may be applied to unbounded loss functions and in particular to the negative log-likelihood (NLL) loss [Bousquet and Elisseeff, 2002, Rakhlin et al., 2005, Shalev-Shwartz et al., 2009, Hardt et al., 2015, Zhang et al., 2016]. However, stability bounds for convex loss functions, e.g., for logistic regression, do not apply to deep nets since they require the NLL loss to be a convex function of the parameters. Alternatively, Hardt et al. [2015] and Kuzborskij and Lampert [2017] estimate the stability of stochastic gradient descent dynamics, which strongly relies on early stopping. This approach results in weaker bounds for the non-convex setting. Stability PAC-Bayesian bounds for bounded and Lipschitz loss functions were developed by London [2017]. Li et al. [2019] utilized the log-Sobolev inequality to bound the KL divergence under the PAC-Bayesian setting while assuming a bounded loss function. In contrast, we assume an unbounded loss function and a Gaussian prior distribution over the model weights. The latter allows us to compute the KL divergence. Holland [2019] studies PAC-Bayesian learning guarantees for heavy-tailed losses. Haddouche et al. [2021] relax the boundness assumption by modifying the range of the loss to depend on each predictor. Differently, through the log-Sobolev inequality, our bound relies on the expected gradient-norm of the loss function, which may be unbounded as well.

PAC-Bayesian bounds were recently applied to deep nets [McAllester, 2013, Dziugaite and Roy, 2017, Neyshabur et al., 2017, Pérez-Ortiz et al., 2021]. In contrast to our work, those related works all consider bounded loss functions. An excellent survey on PAC-Bayesian bounds was provided by Germain et al. [2016]. Alquier et al. [2016] introduced PAC-Bayesian bounds for linear classifiers trained with a hinge-loss by explicitly bounding its moment generating function. Dziugaite et al. [2021] investigate the role of data in learning a PAC-Bayesian prior. Closely, Foong et al. [2021] study the tightness of PAC-Bayesian bounds for small datasets. Alquier et al. [2012] provide an analysis for PAC-Bayesian bounds with Lipschitz functions. Our work differs as we derive PAC-Bayesian bounds for non-Lipschitz functions. Work by Germain et al. [2016] is closer to our setting and considers PAC-Bayesian bounds for linear models and quadratic loss functions. In contrast, our work considers the multi-class setting and non-linear models. PAC-Bayesian bounds for the NLL loss in the online setting were put forward too [Takimoto and Warmuth, 2000, Banerjee, 2006, Bartlett et al., 2013, Grünwald and Mehta, 2017]. The online setting does not consider the whole sample space and therefore is simpler to analyze in the Bayesian setting. Recently, a PAC-Bayesian generalization bound for meta-learning was proposed [Rothfuss et al., 2020, Amit and Meir, 2018, Farid and

Majumdar, 2021, Ding et al., 2021]. In contrast to our work, Amit and Meir [2018] and Farid and Majumdar [2021] focus on bounded loss functions while Rothfuss et al. [2020] assume the loss function to be sub-gamma.

PAC-Bayesian bounds for the NLL loss function are intimately related to learning Bayesian inference Germain et al. [2016]. Recently, many works applied various posteriors in Bayesian deep nets. Gal and Ghahramani [2015], Gal [2016] introduce a Bayesian inference approximation using Monte Carlo (MC) dropout, which approximates a Gaussian posterior using Bernoulli dropout. Srivastava et al. [2014], Kingma et al. [2015] introduced Gaussian dropout, which effectively created a Gaussian posterior that couples the mean and the variance of the learned parameters and explored the relevant log-uniform priors. Blundell et al. [2015], Louizos and Welling [2016] suggest taking a complete Bayesian perspective and learning the mean and the variance of each parameter separately.

Stochastic gradient Langevin dynamics (SGLD) bounds [Pensia et al., 2018, Mou et al., 2018, Negrea et al., 2019, Haghifam et al., 2020] suggest bounding the generalization gap by investigating the dynamics of the parameters through Langevin dynamics and Fokker-Planck equations. Broadly, the Fokker-Planck equation defines a Markov process, and these works measure the KL-divergence between a prior and posterior distribution of this process in the parameter space using gradient norms of the parameters. Differently, our work utilizes the Herbst theorem to measure the KL-divergence (in the form of functional entropy) between prior and posterior in the input space. This distinction leads to a computation of the gradients with respect to different quantities, i.e., parameters vs. input data.

# 6 Discussion and future work

In this study, we present new PAC-Bayesian generalization bounds that depend on the gradient-norm of the loss function. We explore their properties in our experimental validation in various deep learning settings, reinforcing the importance of gradients in contemporary deep nets. Our framework allows for new PAC-Bayesian bounds that assume an on-average bounded loss and an on-average bounded gradient norm assumption. These findings answer an open problem raised by Bartlett et al. [2017a] under the PAC-Bayesian setting. Moreover, we extend the current generalization bounds proposed for the hinge-loss to any linear model with a Lipschitz loss function. Finally, we empirically show that our bounds produce tighter generalization performance than the baseline methods. These results are another step towards bridging the gap between theory and practice while having more realistic assumptions for modern deep nets.

Our framework can be extended in different directions. Log-Sobolev inequalities (LSIs) are intimately related to hypercontractivity. Our generalization bounds also imply that deep nets hyper-contracts their input and therefore generalize. While the relation between generalization and hypercontractivity is under explored, with the small-ball theorem as a notable exception [Lecué and Mendelson, 2017, Mendelson, 2014], we believe it is a fundamental concept in contemporary deep nets that require further research. Additionally, our framework focuses on bounding the complexity term. It is interesting future work to also consider the KL term.

We assume that the labels are balanced with respect to the loss, or equivalently, the influence of all labels is equal. The theory of influence of discrete functions is well-developed and very relevant to LSIs, hypercontractivity, and measure concentration, and further investigation on its relation to generalization is desired [O'Donnell, 2014].

**Acknowledgements:** This work is supported in part by NSF under Grant # 2008387, and, BSF under Grant # 2019783.

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
