# Appendix - On the Importance of Gradient Norm in PAC-Bayesian Bounds

Itai Gat[1], Yossi Adi[2,3], Alexander Schwing[4], Tamir Hazan[1]

[1] Technion
[2] FAIR Team, Meta AI Research
[3] The Hebrew University of Jerusalem
[4] University of Illinois at Urbana-Champaign

## A Log-Sobolev inequalities

For completeness we derive Theorem 2.3. We begin with proving the LSI for a Rademacher random variable, i.e., a discrete random variable that takes the values $\{-1, +1\}$ with equal probability.

**Theorem A.1** (LSI for Rademacher distribution, Gross [1975]). *let $Z$ be Rademacher random variable that take values in $\{-1, +1\}$ with probability of $\frac{1}{2}$. Set the discrete gradient of a function $f : \{-1, +1\} \to \mathbb{R}$ to be*

$$\nabla f(z) \triangleq \frac{f(z) - f(-z)}{2}. \tag{1}$$

*Then the following LSI holds:*

$$\text{Ent}\left[e^{f(Z)}\right] \leq 2\,\mathbb{E}_{Z \sim \{-1, +1\}}\left[\left(\nabla e^{f(Z)/2}\right)^2\right]. \tag{2}$$

*Proof.* The quantities that constitute the LSI for Rademacher random variables take the form:

$$\text{Ent}\left[e^{f(Z)}\right] = \frac{e^{f(1)}f(1) + e^{f(-1)}f(-1)}{2} - \frac{e^{f(1)} + e^{f(-1)}}{2} \log\left(\frac{e^{f(1)} + e^{f(-1)}}{2}\right)$$

$$\mathbb{E}_{Z \sim \{-1, +1\}}\left[\left(\nabla e^{f(Z)/2}\right)^2\right] = \left(\frac{e^{f(1)/2} - e^{f(-1)/2}}{2}\right)^2. \tag{3}$$

Setting $a = f(1)$ and $b = f(-1)$ the log-Sobolev inequality implies that:

$$\frac{1}{2}(a^2 \log a^2 + b^2 \log b^2) - \frac{a^2 + b^2}{2} \log \frac{a^2 + b^2}{2} \leq \frac{1}{2}(b - a)^2. \tag{4}$$

In the following, we assume without loss of generality that $a \leq b$. Let's also define

$$g(r) \triangleq \frac{1}{2}(a^2 \log a^2 + r^2 \log r^2) - \frac{a^2 + r^2}{2} \log \frac{a^2 + r^2}{2} - \frac{1}{2}(r - a)^2. \tag{5}$$

Given this notation, proving the LSI for Rademacher random variables amounts to proving that $g(b) \leq 0$. To prove the inequality $g(b) \leq 0$ we follow these steps:

1. $g'(a) = 0$,

2. for any $a < r \leq b$ holds $g'(r) \leq 0$.

36th Conference on Neural Information Processing Systems (NeurIPS 2022).

First, we have:

$$g'(r) = r \log r^2 + r - r \left( \log \left( \frac{a^2 + r^2}{2} \right) + 1 \right) - (r - a) \tag{6}$$

$$= r \log r^2 - r \log \left( \frac{a^2 + r^2}{2} \right) - (r - a). \tag{7}$$

One can easily verify that $g'(a) = 0$. To prove that $g'(r) \leq 0$ we show that $g''(r) \leq 0$ for any $a < r \leq b$, thus guaranteeing that $g'(r)$ is non-positive.

$$g''(r) = \log r^2 + 2 - \log \left( \frac{a^2 + r^2}{2} \right) - \frac{2r^2}{a^2 + r^2} - 1 \tag{8}$$

$$= \log \left( \frac{2r^2}{a^2 + r^2} \right) - \frac{2r^2}{a^2 + r^2} + 1. \tag{9}$$

Using the fact that $1 + \log \alpha - \alpha \leq 0$ for any $\alpha$ we are able to verify that $g''(r) \leq 0$. □

The LSI for Rademacher complexity extends to Gaussian distributions using the central limit theorem, which asserts that an average of i.i.d. Rademacher random variables converges to Gaussian distribution. This extension relies on tensorization of the functional entropy, which effortlessly scales to high-dimensional settings.

**Theorem A.2** (Tensorization of Rademacher random variables, Gross [1975], Boucheron et al. [2013]). *Let $Z_1, ..., Z_d$ be i.i.d. Rademacher random variables and let*

$$\text{Ent}_i[f(z_1, ..., z_d)] \triangleq \text{Ent}[f(z_1, ..., z_{i-1}, Z_i, z_{i+1}, ..., z_d)] \tag{10}$$

$$\triangleq \mathbb{E}_{Z_i}[f(z_1, ..., z_{i-1}, Z_i, z_{i+1}, ..., z_d) \log(f(z_1, ..., z_{i-1}, Z_i, z_{i+1}, ..., z_d))] \tag{11}$$
$$- (\mathbb{E}_{Z_i}[f(z_1, ..., z_{i-1}, Z_i, z_{i+1}, ..., z_d)]) \log (\mathbb{E}_{Z_i}[f(z_1, ..., z_{i-1}, Z_i, z_{i+1}, ..., z_d)]).$$

*Tensorization of independent random variables $Z_1, ..., Z_d$ implies the bound:*

$$\text{Ent}[f(Z_1, ..., Z_d)] \leq \mathbb{E}_{Z_1, ..., Z_d}[\sum_{i=1}^{d} \text{Ent}_i[f(Z_1, ..., Z_d)]]. \tag{12}$$

*Proof.* We denote the $d$-th tuple by $z = (z_1, ..., z_d)$ and set $z_{[d] \backslash i} = (z_1, ..., z_{i-1}, z_{i+1}, ..., z_d)$. Following the definitions:

$$\text{Ent}[f(Z_1, ..., Z_d)] = 2^{-d} \sum_z f(z) \log f(z) - \left( 2^{-d} \sum_z f(z) \right) \log \left( 2^{-d} \sum_z f(z) \right), \tag{13}$$

$$\text{Ent}_i[f(z_1, ..., z_d)] = 2^{-1} \sum_{z_i} f(z) \log(f(z)) - \left( 2^{-1} \sum_{z_i} f(z) \right) \log \left( 2^{-1} \sum_{z_i} f(z) \right). \tag{14}$$

$$\tag{15}$$

We assume without loss of generality[1] that $\sum_z 2^{-d} f(z) = 1$ and set $q(z) \triangleq 2^{-d} f(z)$. Since $f(z) \geq 0$, then $q(z)$ is a distribution and its entropy is $H(q) = -\sum_z q(z) \log q(z)$. Then:

$$\text{Ent}[f(Z_1, ..., Z_d)] = d - H(q). \tag{16}$$

We denote the marginal distribution by $q(z_{[d] \backslash i}) = \sum_{z_i} q(z_{[d] \backslash i}, z_i)$. Then

$$\text{Ent}_i[f(z_1, ..., z_d)] = \sum_{z_i} q(z_{[d] \backslash i}, z_i) \log(q(z_{[d] \backslash i}, z_i)) - q(z_{[d] \backslash i}) \log(q(z_{[d] \backslash i})) \tag{17}$$

$$\mathbb{E}_{Z_1, ..., Z_d}[\text{Ent}_i[f(z_1, ..., z_d)]] = -H(q) + H(q_{[d] \backslash i}) + 1, \tag{18}$$

---

[1]Since $\text{Ent}(cf) = c \, \text{Ent}(f)$, for any $c > 0$, then $\text{Ent}[cf] \leq \mathbb{E}[\sum_{i=1}^{d} \text{Ent}_i[cf]]$ if and only if $\text{Ent}[f] \leq \mathbb{E}[\sum_{i=1}^{d} \text{Ent}_i[f]]$.

where $q_{[d]\setminus i}$ is the distribution of $q(z_{[d]\setminus i})$. With these equalities in mind,

$$\text{Ent}[f(Z_1,...,Z_d)] \leq \mathbb{E}_{Z_1,...,Z_d}\left[\sum_{i=1}^{d}\text{Ent}_i[f(z_1,...,z_d)]\right] \iff H(q) \leq \frac{1}{d-1}\sum_{i=1}^{d}H(q_{[d]\setminus i}) \quad \text{(Han's inequality)}. \quad (19)$$

The right hand side of the above equation is know as Han's inequality.

$\square$

We use tensorization and the central limit theorem to extend the LSI of Rademacher random variables to derive the LSI for Gaussian random variables.

**Theorem A.3** (Gaussian log-Sobolev inequality (LSI), Gross [1975]). *Let $Z$ be a Gaussian random variables $Z \sim \mathcal{N}(0,1)$. Then*

$$\text{Ent}[e^{f(Z)}] \leq \frac{1}{2}\mathbb{E}[f'(Z)^2 e^{f(Z)}]. \quad (20)$$

*Proof.* In the following, we denote by $Z_1,...,Z_d$ Rademacher random variables. We set $f_d(Z_1,...,Z_d) = f\left(\frac{\sum_{i=1}^{d}Z_i}{\sqrt{d}}\right)$. Using tensorization and the LSI for Rademacher random variables:

$$\text{Ent}[f_d(Z_1,...,Z_d)] \quad \leq \quad \mathbb{E}_{Z_1,...,Z_d}[\sum_{i=1}^{d}\text{Ent}_i[f_d(Z_1,...,Z_d)]] \quad (21)$$

$$\leq \quad \frac{1}{2}\mathbb{E}_{Z_1,...,Z_d}\left[\sum_{i=1}^{d}\mathbb{E}_{Z_i\sim\{-1,+1\}}\left[\left(\frac{f\left(\frac{\sum_{j=1}^{d}Z_j}{\sqrt{d}}\right) - f\left(\frac{\sum_{j\neq i}^{d}Z_j}{\sqrt{d}} - \frac{Z_i}{\sqrt{d}}\right)}{2}\right)^2 e^{f\left(\frac{\sum_{i=1}^{d}Z_i}{\sqrt{d}}\right)}\right]\right] \quad (22)$$

$$= \quad \frac{1}{2}\mathbb{E}_{Z_1,...,Z_d}\left[\frac{1}{d}\sum_{i=1}^{d}\mathbb{E}_{Z_i\sim\{-1,+1\}}\left[\left(\frac{f\left(\frac{\sum_{j=1}^{d}Z_j}{\sqrt{d}}\right) - f\left(\frac{\sum_{j\neq i}^{d}Z_j}{\sqrt{d}} - \frac{Z_i}{\sqrt{d}}\right)}{\frac{2}{\sqrt{d}}}\right)^2 e^{f\left(\frac{\sum_{i=1}^{d}Z_i}{\sqrt{d}}\right)}\right]\right] \quad (23)$$

The theorem follows using the central limit theorem as $\lim_{d\to\infty}\frac{\sum_{i=1}^{d}Z_i}{\sqrt{d}} = Z$, when $Z \sim \mathcal{N}(0,1)$ and

$$f(Z) \quad = \quad \lim_{d\to\infty}f_d(Z_1,...,Z_d), \quad (24)$$

$$f'(Z) \quad = \quad \lim_{d\to\infty}\frac{f\left(\frac{\sum_{j=1}^{d}Z_j}{\sqrt{d}}\right) - f\left(\frac{\sum_{j\neq i}^{d}Z_j}{\sqrt{d}} - \frac{Z_i}{\sqrt{d}}\right)}{\frac{2}{\sqrt{d}}}. \quad (25)$$

$\square$

The above provides a LSI for the standard Gaussian distribution. While we can use the same technique to prove a LSI for $Z \sim \mathcal{N}(\mu,\sigma^2)$, we separate these two theorems for clarity and turn to prove the general case as a corollary.

**Theorem A.4** (Gaussian log-Sobolev inequality (LSI), Gross [1975]). *Let $Z \sim \mathcal{N}(\mu,\sigma^2)$ be a Gaussian random variable with mean $\mu$ and variance $\sigma^2$. Then*

$$\text{Ent}[e^{f(Z)}] \leq \frac{1}{2}\sigma^2\,\mathbb{E}[f'(Z)^2 e^{f(Z)}]. \quad (26)$$

*Proof.* The proof follows via a simple change of variable. Let $\hat{Z} \sim \mathcal{N}(0,1)$, then $\mu+\sigma\hat{Z} \sim \mathcal{N}(\mu,\sigma^2)$. We set $g(\hat{Z}) \triangleq f(\mu+\sigma Z)$. We use the LSI for the standard Gaussian: $\text{Ent}[e^{g(\hat{Z})}] \leq \frac{1}{2}\mathbb{E}[g'(\hat{Z})^2 e^{g(\hat{Z})}]$ and note that $g'(\hat{z}) = \sigma f'(z)$. $\square$

Theorem 2.3 for multivariate Gaussians holds when applying tensorization for each Gaussian independently.

# B Proofs

**Theorem B.1.** *Let $\ell(w, x, y) \triangleq \hat{\ell}(Wx, y)$ be a differentiable loss function over $x = (x_1, ..., x_d)$ and $k$ classes $y \in \{1, ..., k\}$, with Lipschitz constant $g$, i.e., $\|\nabla_t \hat{\ell}(t, y)\| \leq g$. Consider a Gaussian prior distribution $p \sim N(0, \sigma_p^2)$. Under the conditions of Lemma 3.3, for any $0 < \lambda \leq \sqrt{\frac{m}{16}}/g\sigma_p\sigma_y$ and $\delta \in (0, 1)$, with probability at least $1 - \delta$ over the draw of the training set $S$, we have*

$$\mathbb{E}_{w \sim q}[L_D(w)] \leq \mathbb{E}_{w \sim q}[L_S(w)] + \frac{kd\log(\sqrt{4/3}) + KL(q||p) + \log(1/\delta)}{\lambda}. \tag{27}$$

*Proof.* This bound is derived by applying Lemma 3.3. We begin by realizing the gradient of $\hat{\ell}(Wx, y)$ with respect to $x$. Using the chain rule, $\nabla_x \hat{\ell}(Wx, y) = W^\top \nabla_{Wx} \hat{\ell}(Wx, y)$. Hence, we obtain for the gradient norm $\|\nabla_x \hat{\ell}(Wx, y)\|^2 \leq \|\nabla_{Wx} \hat{\ell}(Wx, y)\|^2 \cdot \sum_{y=1}^{k} \sum_{j=1}^{d} w_{y,j}^2 \leq g^2 \sum_{y=1}^{k} \sum_{j=1}^{d} w_{y,j}^2$. Plugging this result into Lemma 3.3, we obtain the following bound:

$$\mathbb{E}_{\mathcal{D}} \left[ \|\sigma_y \nabla \ell(w, x, y)\|^2 \int_0^{\frac{\lambda}{m}} \frac{e^{-\alpha \ell(w,x,y)}}{M(\alpha)} d\alpha \right] \leq \sigma_y^2 g^2 \sum_{y=1}^{k} \sum_{j=1}^{d} w_{y,j}^2 \cdot \mathbb{E}_{\mathcal{D}} \left[ \int_0^{\frac{\lambda}{m}} \frac{e^{-\alpha \ell(w,x,y)}}{M(\alpha)} d\alpha \right]$$

$$= \sigma_y^2 g^2 \sum_{y=1}^{k} \sum_{j=1}^{d} w_{y,j}^2 \int_0^{\frac{\lambda}{m}} \frac{\mathbb{E}_{\mathcal{D}} \left[ e^{-\alpha \ell(w,x,y)} \right]}{M(\alpha)} d\alpha.$$

Since $M(\alpha) \triangleq \mathbb{E}_{\mathcal{D}} \left[ e^{-\alpha \ell(w,x,y)} \right]$, the ratio in the integral equals one and the integral $\int_0^{\frac{\lambda}{m}} d\alpha = \frac{\lambda}{m}$. Combining these results we obtain:

$$C(\lambda, p) \leq \log \left( \mathbb{E}_{w \sim p} e^{\frac{\sigma_y^2 \lambda^2 g^2}{2m} \Sigma_{y,j} w_{y,j}^2} \right). \tag{28}$$

Finally, whenever $\lambda g \sigma_p \sigma_y \leq \sqrt{m/4}$ we follow the Gaussian integral and derive the bound

$$\log \left( \mathbb{E}_{w \sim p} e^{\frac{\sigma_y^2 \lambda^2 g^2}{2m} \Sigma_{y,j} w_{y,j}^2} \right) = \log \left( \sqrt{\frac{m}{m - 4\sigma_y^2 \lambda^2 g^2 \sigma_p^2}} \right)^{kd} \tag{29}$$

$$\leq kd \cdot \log \left( \sqrt{\frac{m}{m - m/4}} \right) = kd \cdot \log(\sqrt{4/3}). \tag{30}$$

Finally, we obtain Equation (27) by plugging the above into Theorem 2.1. $\qquad \square$

**Theorem B.2.** *Consider smooth loss functions that are on-average bounded, i.e., for every $w$ the following holds: $\mathbb{E}_{\mathcal{D}} \ell(w, x, y) \leq b$ and $\mathbb{E}_{\mathcal{D}} \left[ \|\nabla_x \ell(w, x, y)\|^2 \right] \leq g$. Under the conditions of Lemma 3.3 for any $0 < \lambda \leq m$ and $\delta \in (0, 1)$, with probability at least $1 - \delta$ over the draw of the training set $S$, we obtain*

$$\mathbb{E}_{w \sim q}[L_D(w)] \leq \mathbb{E}_{w \sim q}[L_S(w)] + \frac{\frac{\lambda^2 e^b g \sigma_y^2}{2m} + KL(q||p) + \log(1/\delta)]}{\lambda}. \tag{31}$$

*Proof.* This bound is derived by applying Lemma 3.3 and bounding $\int_0^1 \frac{e^{-\alpha \ell(w,x,y)}}{M(\alpha)} d\alpha \leq e^b$. We derive this bound in three steps: First, from $\ell(w, x, y) \geq 0$ we obtain

$$e^{-\alpha \ell(x,x,y)} \leq 1. \tag{32}$$

Then, we lower bound $M(\alpha) \geq M(1)$ for any $0 \leq \alpha \leq \lambda/m$: we note that $0 < \lambda \leq m$, therefore we consider $0 \leq \alpha \leq 1$. Also, since $\ell(w, x, y) \geq 0$ the function $e^{-\alpha \ell(w,x,y)}$ is monotone in $\alpha$ within the unit interval, i.e., for $0 \leq \alpha_1 \leq \alpha_2 \leq 1$ there holds

$$e^{-\alpha_1 \ell(w,x,y)} \geq e^{-\alpha_2 \ell(w,x,y)} \tag{33}$$

and consequently $M(\alpha) \geq M(1)$ for any $\alpha \leq 1$. Lastly, the assumption $\mathbb{E}_{\mathcal{D}}[-\ell(w, x, y)] \geq -b$ and the monotonicity of the exponential function result in the lower bound

$$e^{\mathbb{E}_{\mathcal{D}}[-\ell(w,x,y)]} \geq e^{-b}. \tag{34}$$

From convexity of the exponential function, $M(1) = \mathbb{E}_{\mathcal{D}}\, e^{-\ell(w,x,y)} \geq e^{\mathbb{E}_{\mathcal{D}}[-\ell(w,x,y)]}$, and the lower bound $M(1) \geq e^{-b}$ follows.

Combining these bounds we derive the upper bound

$$\int_0^1 \frac{e^{-\alpha\ell(w,x,y)}}{M(\alpha)}d\alpha \leq \int_0^1 \frac{1}{e^{-b}}d\alpha = e^b, \tag{35}$$

and the result follows.

Next, by replacing $\mathbb{E}_{\mathcal{D}}\left[\|\sigma_y\nabla\ell(w,x,y)\|^2\right]$ with $g\sigma_y^2$, we obtain

$$C(\lambda, p) \leq \log\mathbb{E}_{w\sim p}\, e^{\frac{1}{2}\lambda\,\mathbb{E}_{\mathcal{D}}\left[\|\sigma_y\nabla\ell(w,x,y)\|^2 \int_0^{\frac{\lambda}{m}} \frac{e^{-\alpha\ell(w,x,y)}}{M(\alpha)}d\alpha\right]}$$

$$\leq \log\mathbb{E}_{w\sim p}\, e^{\frac{\lambda^2 e^b g\sigma_y^2}{2m}} = \frac{\lambda^2 e^b g\sigma_y^2}{2m}. \tag{36}$$

Finally, we obtain Equation (31) by plugging Equation (36) into Theorem 2.1. $\square$

## C   Additional results

### C.1   Connection between generalization and bound

To further demonstrate the use of the bound, we performed a new experiment. We study the effect of important components in ResNet using our bound. For this, we train four variations of the ResNet18 model: 1) a standard model (ResNet); 2) a model without skip connections (ResNetNoSkip); 3) a model without batch normalization layers (ResNetNoBN); and 4) a model without both skip connections and batch normalization layers (ResNetNoSkipNoBN). We optimize all models on the CIFAR10 data: We observe that ResNet and ResNetNoSkip achieve comparable performance in all

| MODEL | TEST LOSS | TRAIN LOSS | BOUND ON $C(\sqrt{m}, p)$ |
|---|---|---|---|
| RESNET | $0.722\pm0.01$ | $0.541\pm0.06$ | $0.185\pm0.006$ |
| RESNETNOSKIP | $0.631\pm0.02$ | $0.478\pm0.05$ | $0.179\pm0.009$ |
| RESNETNOBN | $0.603\pm0.05$ | $0.564\pm0.08$ | $0.172\pm0.007$ |
| RESNETNOSKIPNOBN | $2.302\pm0.01$ | $2.31\pm0.02$ | $0.01\pm0.0004$ |

metrics. Additionally, removing the batch normalization layers and including the skip connections achieves comparable performance to ResNet and ResNetNoSkip. Similar to Zhang et al. [2019], this finding suggests that even without batch normalization, models can converge using precise initialization. Interestingly, by removing batch normalization and skip connection layers, the model gets to a rate of $\lambda = m$ and achieves a good generalization bound. However, this comes at the expense of poor model fitting to the train set due to gradient vanishing. These results are consistent with prior findings in which batch normalization improves optimization [Santurkar et al., 2018]. To conclude, we were able to obtain a tight upper bound of the generalization gap with our proposed bound. However, it is important to note that when using any generalization bound, one should care about the training loss as well as the complexity term.

### C.2   Gradient statistics

We further study the gradient norm statistics, we report the max and mean values of the gradient norm (not squared) using the same networks described in Section. 4 in the main paper:

| # LAYERS | MNIST (MEAN) | MNIST (MAX) | CIFAR (MEAN) | CIFAR (MAX) |
|---|---|---|---|---|
| 1 | 0.00224 | 0.00267 | 0.0258 | 0.0314 |
| 2 | 0.00088 | 0.0012 | 0.011 | 0.015 |
| 3 | 0.00036 | 0.00056 | 0.0049 | 0.008 |

We observe that the maximum value of the gradient norm is not higher than twice the mean value.

# D   Experimental setup

For all MNIST experiments we use MLPs of depth $d \in \{1, \ldots, 5\}$. For CIFAR10 experiments we use CNNs of depth $d \in \{2, \ldots, 5\}$. For the CNN models, $d$ denotes the number of convolutional layers. We also added a max-pooling layer after each convolutional layer. We included two additional fully connected layers in all CNN models to fix the target output dimension. In all models, we use the ReLU activation function. We optimize the negative log-likelihood (NLL) loss function using stochastic gradient descent (SGD) with a learning rate of 0.01 and a momentum value of 0.9 in all settings for 50 epochs. We use mini-batches of size 128 and did not use any learning rate scheduler. To span the possible weights, we sampled from a normal prior distribution with different variances. For a fair comparison, we set the layers' width to reach roughly the same number of parameters in each model (except for the linear case). All reported results use $\delta = 0.01$.