# OpenReview forum: "On the Importance of Gradient Norm in PAC-Bayesian Bounds"
_NeurIPS.cc/2022/Conference — NeurIPS 2022 Accept_

### Official Review · Reviewer_b5Rz · 2022-07-11

**Rating:** 6
**Confidence:** 3
**Soundness:** 3 good
**Presentation:** 3 good
**Contribution:** 3 good

**Summary:**

This paper presented a novel PAC-Bayesian generalization error bound that depends on the gradient norm of the loss function. The idea is to use Herbst argument and control the entropy using the appropriate assumption about data and loss functions and their gradient norm. The obtained bound assumes the average of the loss function and its gradient norm. Numerical experiments suggest that the obtained bound can capture the structure of the deep neural networks.

**Questions:**

- Questions related to Weaknesses.
- The assumption in line 139, input data $x$ follows the Gaussian distribution, is very strong. On the other hand, if we relax such an assumption not the input space but the feature or latent space, then it would be moderate and easier to verify. Is such relaxation possible?

- In lines 233 to 237, the assumption in Lemma 3.3 is evaluated by the numerical experiments. Authors claimed that ``The maximum standard deviation across the labels is 0.022, while the mean value is 4.605. Hence, it is evident that this assumption holds in practice", but I cannot how this conclusion is derived. I would like to know more detail about this point.

**Limitations:**

The limitations (assumptions) are written clearly. Since this paper proposed a theoretical analysis, it seems there exists no social negative impact.

**Strengths And Weaknesses:**

# Strengths
- The motivation of the research is practical and easy to understand.
- As far as I know, using Herbst argument for deriving the generalization error bound seems novel and the assumption of the obtained bound (on average bounded and Lipschitz property of the loss function) is also novel and meaningful.
- The obtained bound naturally includes the multi-class classification, which is practically important.
- The numerical experiments especially Figure 2, can successfully capture the importance of the depth of neural networks in generalization ability.

# Weaknesses
- I think some assumptions are difficult to verify or strong in practice. I list them below.
- To analyze the entropy, the authors assumed that input data $x$ follows the Gaussian distribution (line 139), which is a very strong assumption in practice.
- The assumption in Lemma 3.3, ``the loss per label is balanced'', for any parameter $w$ is very strong and difficult to understand intuitively.

---

> ### Author Response · Authors · 2022-08-02
> **Response to reviewer b5Rz**
>
> **Mixture of Gaussians** - We think the assumptions on the data are not as strong: We assume the data distribution follows a mixture of Gaussians conditioned on the label, i.e., each subset of the data associated with a label is generated from a different mixture model. In addition, please note that the number of components in the mixture model is not limited. Such a Gaussian mixture model can approximate smooth densities [1, 2, 3]. Hence, we don’t consider this data assumption to be strict.
>
> The root cause of this assumption is the use of the log-Sobolev inequality. It is possible to prove the log-Sobolev inequality for a general probability measure by making more assumptions about the loss function.
>
> Based on the implications of more restricted assumptions on the loss function and the data, we used a mixture of Gaussians. We agree with the reviewer that it is an important discussion, and we will add it to the camera ready.
>
> [1] Statistical Analysis of Finite Mixture Distributions, D. M. Titterington, A. F. M. Smith, and H. E. Makov, 1985
>
> [2] Multivariate Density Estimation: Theory, Practice, and Visualization, David W. Scott, 1992
>
> [3] Deep Learning, Ian Goodfellow and Yoshua Bengio, and Aaron Courville, 2016
>
> **Per-label loss balance** - The per-label loss balance originates from bounding the term $\text{Ent}_{\mathcal{D}_y}[\mathbb{E}\_{x\sim \mathcal{N}_y}[f_w]]$ in Eq. 17. If the loss is per-label balanced, the entropy equals zero. In fact, the per-label balance can be relaxed by assuming that the loss is per-label bounded within an amplitude (See 3.16 in [4]). Please also note that we assume the per-label balance assumption is based on the prior distribution of the parameters. We decided not to present this relaxation in the paper as it complicates the theorems, and as we discuss next, the per-label balance holds in practice.
>
> Please also note that we empirically verify this assumption in Sec. 4 (L233-237). We conducted the following experiment: given a classifier and a dataset, we evaluated the loss value for each label. Using ten different architectures and two datasets, the mean value of the loss of each label was 4.605, and the standard deviation was 0.022. This indicates that the per-label balance assumption holds, as the loss values do not change much across labels. We agree with the reviewer that it might be confusing. Thank you for the suggestion. We will clarify it in the camera-ready version.
>
> [4] Probability in High Dimension, Ramon van Handel, 2016

---

> ### Comment · Area_Chair_G85Q · 2022-08-08
> **please acknowledge the authors' response**
>
> Please acknowledge the authors' response.

---

### Official Review · Reviewer_RE5w · 2022-07-11

**Rating:** 6
**Confidence:** 4
**Soundness:** 3 good
**Presentation:** 3 good
**Contribution:** 3 good

**Summary:**

The paper extends PAC-Bayes generalization literature by proving new generalization bounds. These bounds require (and depend on) the expected loss (and loss gradient norm) to be bounded, where the expectation is taken over the data distribution. In general, these assumptions on the bound on the loss and gradient have to be valid uniformly over the weight space, or when integrated over the prior on the weights (if computing such a quantity is possible). The bounds were derived for a data distribution that is a mixture of Gaussians. The authors also include an empirical section, measuring how the complexity terms appearing in the bounds change with a parameter lambda that appears in the bounds, as well as the depth of the network.

**Questions:**

Related work issues: the literature on information-theoretic approaches to generalization contains several papers that depend on gradient norms, an average deviations from the norms. The paper under submission is obviously related, and thus I believe a deeper comparison is needed. The authors should check a fairly large body of work, such as Negrea et al ‘19, Haghifam et al ‘20, Mou et al. ‘18, Pensia et al. ‘18.

Empirical evaluation / Section 4:
Section 4 contains a paragraph on verifying assumptions, but there is no mention of the assumption that the data is sampled from a mixture of Gaussians. This is the key assumption used to derive the main result.
There is no plot directly comparing average gradient versus max gradient, which is the key distinction.
Fig 1 shows how the expected loss and loss gradient vary with sigma (prior variance). It looks like these quantities are reported at a fixed w (the one learned by SGD?). If so, the relevant quantity is the one where w is also random, and sampled from the posterior, and the expectation should be taken over w.
Given the issue with Figure 1, I would also like the authors to explicitly confirm that Table 1 reports average error of randomized classifiers.

When discussing subgaussianity in Sec 2.1, and assumptions for Herbst theorem in Section 2.2,  the same letter sigma is being used, despite sigma referring to two different quantities (if i understand correctly). I found it a bit confusing at first, and wondered for some time whether 2.2 uses subgaussianity  in any way.

Theorem 2.2 gives a bound on the log moment generating function that essentially looks like a bound for subgaussian random variables, but also has an extra term E[F]. For a subgaussian loss, what would sigma in Eq (5) look like?

It seems like Theorem 2.3 could have been stated for a single draw from a gaussian, given it’s use in the proof of Lemma 3.3.


The proofs for existing results (that are not derived by the authors) are included in the main text, but the proofs for new results are all in the appendix. I think the authors should at least include the outlines in the main text, and, if needed, shorten the other proofs.

Other comments and minor issues:
Notation in theorem 3.4 is unclear: does one need a bound on the loss gradient norm to hold uniformly over y? Or t (so w, and x, i suppose)?
Theorem 3.5 what is the gradient w.r.t.?
Line 15 “they do not tend to overfit” is not really accurate, depends on the exact definition of overfitting, what loss overfitting is being measured w.r.t., etc. The same holds for ~line 271.
Lines 100, 253 typos
What do the authors mean by “avoid integrating” in line 166?
- line 333 typo, lower case log.

**Limitations:**

The authors discuss how their work can be extended. However, there is no explicit discussion of limitations. I think the major one if the assumption on the data distribution (mixture of gaussians). This should be mentioned explicitly in the abstract, as well as discussed under limitations. I personally do not see any societal impact of this work beyond standard impact of theory and better understanding of deep learning, thus for me personally it is not an issue that the authors skipped it.

I believe citations to datasets and architectures used are missing.

**Strengths And Weaknesses:**

Overall, it’s a pretty good and interesting paper, and I believe the advancements in PAC-Bayes theory are worth a publication and novel. I like that the authors did not only stick to theory, but also evaluated their bound empirically. However, there are a few issues
 - with the experiments (i hope the authors will answer my questions, and confirm the correctness of their evaluation of the bounds),
 - some presentation issues in the theory part that need to be addressed with care to improve clarity,
 - and also missing references and comparison to prior work, which is the one of the biggest issues for me.

I am leaning towards accepting the paper with the hope that the issues listed above will be addressed by the authors.

---

> ### Author Response · Authors · 2022-08-02
> **Response to reviewer RE5w**
>
> **Prior work** - We thank the reviewer for driving a comparison to stochastic gradient Langevin dynamics (SGLD). This line of work suggests bounding the generalization gap by investigating the dynamics of the parameters $w$ through Langevin dynamics and Fokker-Planck equations. Broadly, the Fokker-Planck equation defines a Markov process, and these works measure the KL-divergence between a prior and posterior distribution of this process (in the parameter space) using gradient norms of the parameters. Differently, our work utilizes the Herbst theorem to measure the KL-divergence (in the form of functional entropy) between prior and posterior in the input space (contrary to the parameter space). This distinction leads to a computation of the gradients with respect to different quantities (parameters vs. input data). We will add those works and this discussion to the camera-ready version.
>
> **Mixture of Gaussians** - We think the assumptions on the data are not as strong: We assume the data distribution follows a mixture of Gaussians conditioned on the label, i.e., each subset of the data associated with a label is generated from a different mixture model. In addition, please note that the number of components in the mixture model is not limited. Such a Gaussian mixture model can approximate smooth densities [1, 2, 3]. Hence, we don’t consider this data assumption to be strict. We agree with the reviewer that it is an important discussion, and we will add it to the camera ready.
>
> [1] Statistical Analysis of Finite Mixture Distributions, D. M. Titterington, A. F. M. Smith, and H. E. Makov, 1985
>
> [2] Multivariate Density Estimation: Theory, Practice, and Visualization, David W. Scott, 1992
>
> [3] Deep Learning, Ian Goodfellow and Yoshua Bengio, and Aaron Courville, 2016
>
> **Mean vs. gradient** - Please note that we report the expected squared gradient norm in Fig. 1 (c-d) as a function of the number of layers and the prior distribution of the weights. To further study the gradient norm statistics, we report the max and mean values of the gradient norm (not squared) using the same networks described in the paper:
>
> |# layers|MNIST (mean)|MNIST (max)|CIFAR (mean)|CIFAR (max)|
> |--|:-:|:-:|:-:|:-:|
> |1|0.00224|0.00267|0.0258|0.0314|
> |2|0.00088|0.0012|0.011|0.015|
> |3|0.00036|0.00056|0.0049|0.008|
>
> We observe that the maximum value of the gradient norm is not higher than twice the mean value. We will add those results to the camera-ready version.
>
> **Figure 1** - Our bound studies the complexity term from a prior weights point of view. Hence, the results in Fig. 1 are indeed computed on randomized classifiers. It is essential to note since our bound works on the prior distribution of the parameters. The expectation of the prior distribution of the parameters can be seen in Theorem 3.6. In retrospect, Theorem 3.6 can be presented clearer. We will clarify it in the camera-ready.
>
> **$\sigma$ in 2.1 and 2.2** - We agree with the reviewer that this notation is confusing. We will modify the subgassian sigma to rho in the camera-ready.
>
> **Herbst** - In L98-99, we connect between entropy bounds and sub-Gaussian measures. The additional term $\mathbb{E}[F]$ does not impact the measure concentration (of a random variable from its mean) as one can always consider the zero-mean random variable $G = F - \mathbb{E}[F]$ that has the same concentration properties ($G$ around mean zero vs. $F$ around mean $\mathbb{E}[F]$).
>
> **Notation in Theorem 2.3 and Lemma 3.3** - We use the notation in Theorem 2.3 to give the reader a better understanding of the tensorization process presented in Theorem A.2 for Rademacher and A.4 for Gaussian. In Lemma 3.3, we use a single high-dimensional Gaussian variable version. We will elaborate on this in the camera-ready version.
>
> **Manuscript recommendation** - Thanks for this suggestion. We will address it while keeping readability in mind.
>
> **Notation in Theorem 3.4** - The bound needs to hold over $y$ for all $Wx$.
>
> **Theorem 3.5 gradient** - The gradient in Theorem 3.5 is taken with respect to the input data $x$.
>
> **Overfit** - we agree with the reviewer that it might be too general, we will modify it in the camera-ready.
>
> **L166** - It is impractical to integrate over $\alpha$ since this term will involve integrals over: parameter prior distribution, data distribution, $\alpha$, and again data distribution (the denominator in the integral over $\alpha$). Later in the paper, we develop Theorems 3.4-3.6, which eliminate the integration over $\alpha$ and the expectation in the denominator accordingly.
>
> **Typos** - Thanks for pointing out, we will fix it in the camera-ready version.
>
> **Limitations** - Thanks for the suggestion. We will provide a more detailed discussion of limitations and discuss data assumptions such as a mixture of Gaussian model as a density approximator (as stated in **Mixture of Gaussians**).
>
> **Citations** - Thanks for the suggestion. We will add missing citations in the camera-ready.

---

> > ### Comment · Reviewer_RE5w · 2022-08-08
> > **Updated draft?**
> >
> > Regarding the mixture of Gaussians assumption, how do your results depend on the number of mixture components?
> >
> > There are a lot of promises in the author response. Note that the draft on openreview can be updated. When should we expect to see a revision?

---

> > > ### Author Response · Authors · 2022-08-08
> > > **Thank you**
> > >
> > > Thanks a lot for reading our rebuttal. We uploaded a revised version of the manuscript. Significant changes are marked in blue. We would be happy to address any remaining concerns in the camera-ready version.
> > >
> > > Regarding the mixture of Gaussians - please note that our results are not dependent on the number of components.

---

### Official Review · Reviewer_2Hhp · 2022-07-11

**Rating:** 6
**Confidence:** 3
**Soundness:** 3 good
**Presentation:** 3 good
**Contribution:** 3 good

**Summary:**

The authors present a new PAC-Bayesian generalization bound in connections with the gradient norm of loss function with respect to the input space. The authors start from the bound presented in Alquier 2016, and compared to the assumption where loss is uniformly bounded, the authors try to give the complexity term a new bound with a more general assumption, i.e. on-average bounded loss and gradient norm. With such assumptions, this new bounds regarding both linear and non-linear models are proved via the Herbst theorem and Log-Sobolev inequalities. Finally, the authors show some empirical study to show the effectiveness of their frame, which can provide tighter generalization bound compared to conventional techniques.


**Questions:**

I hope the authors can address my concerns in their rebuttal..

**Limitations:**

In my opinion, this paper do not involve related issues for its current version.

**Strengths And Weaknesses:**

&nbsp;
### **Strengths**

1. This paper is clear and well written.
2. This paper provides a new generalization bound with a more relaxing assumption. In my opinion, theoretical insights of the generalization bound are valuable for enriching our understanding towards the neural networks. It is encouraged to contribute to this topic. The authors prove it under the assumptions to the on-average bounded loss and gradient norm, which, to my understanding, makes it closer to our practical situations, although the assumption here has not been significantly improved compared to that in Alquier 2016 (i.e. from uniformly bounded loss assumption to on-average bounded loss assumption).
3. It is interesting to see the theoretical connections between the generalization and the gradient norm of loss w.r.t the input space. This may also provide theoretical insights about the model robustness.
4. This paper provides rigorous mathematical proof.

&nbsp;
### **Weaknesses**

1. I think it can be much more better if the authors show the experiments results regarding the connections between the generalization bound, gradient norm and testing performance.
2. From formula 22 and Lemma 3.1 to Lemma 3.3, it seems that the authors exchange the expectation and integral operation. I am somehow worried about whether such exchange is eligible.
2. Formula 8, $Ent[e^{\alpha F]}$ -> $Ent[e^{\alpha F}]$.
3. Formula 23, it seems that the authors are missing the explanation of $d$, which may be $W \in R^{k \times d}$.

---

> ### Author Response · Authors · 2022-08-02
> **Response to reviewer 2Hhp**
>
> **More models** - Please note Fig. 2c, where we study our bound of the complexity term as a function of the network depth of MLP and CNN models using CIFAR10 and MNIST data. The complexity term decreases as the network depth increases. Also, we present the complexity values for different rates of \lambda in Fig. 2 (a-b). Please also note that we report statistics for the per-label balance assumption in Sec. 4 (L233-237).
>
> To further demonstrate the use of the bound, we performed a new experiment. We study the effect of important components in ResNet using our bound. For this, we train four variations of the ResNet18 model: 1) a standard model (ResNet); 2) a model without skip connections (ResNetNoSkip); 3) a model without batch normalization layers (ResNetNoBN); and 4) a model without both skip connections and batch normalization layers (ResNetNoSkipNoBN). We optimize all models on the CIFAR10 data:
>
> |Model|Test loss|Train loss|Bound on $C(\sqrt{m}, p)$|
> |--|:--:|:--:|:--:|
> |ResNet|0.722$\pm$0.01|0.541$\pm$0.06|0.185$\pm$0.006|
> |ResNetNoSkip|0.631$\pm$0.02|0.478$\pm$0.05|0.179$\pm$0.009|
> |ResNetNoBN|0.603$\pm$0.05|0.564$\pm$0.08|0.172$\pm$0.007|
> |ResNetNoSkipNoBN|2.302$\pm$0.01|2.31$\pm$0.02|0.01$\pm$0.0004|
>
> We observe that ResNet and ResNetNoSkip achieve comparable performance in all metrics. Additionally, removing the batch normalization layers and including the skip connections achieves comparable performance to ResNet and ResNetNoSkip. Similar to Zhang et al. [5], this finding suggests that even without batch normalization, models can converge using precise initialization. Interestingly, by removing batch normalization and skip connection layers, the model gets to a rate of \lambda=m and achieves a good generalization bound. However, this comes at the expense of poor model fitting to the train set due to gradient vanishing. These results are consistent with prior findings in which batch normalization improves optimization [6]. To conclude, we were able to obtain a tight upper bound of the generalization gap with our proposed bound. However, it is important to note that when using any generalization bound, one should care about the training loss as well as the complexity term.
>
> [5] Fixup Initialization: Residual Learning Without Normalization, Hongyi Zhang and Yann N. Dauphin and Tengyu Ma, 2019
>
> [6] How does batch normalization help optimization?, Santurkar, Shibani and Tsipras, Dimitris and Ilyas, Andrew and Madry, Aleksander, 2018
>
> **Integral and expectation** - This exchange comes from Fubini’s theorem. If the double integral yields a finite result after replacing the integrand by its absolute value, Fubini’s theorem allows us to switch the order of integration. In our case, we switched between the integral over $\alpha$ and the data distribution.
>
> **Typo** - Thanks, we will modify it in the camera-ready version.
>
> **Eq. 23** - d is the dimension of the input data x. It is defined in L138. We will clarify.

---

> > ### Comment · Reviewer_2Hhp · 2022-08-07
> > **Thank the author for the response.**
> >
> > Thank the author for the kind responses and interpretations. And I appreciate the additional results wrt other models. But this is not the core intention of my concerns.

---

> > > ### Author Response · Authors · 2022-08-08
> > > **Thank you**
> > >
> > > Thanks a lot for reading our rebuttal. We certainly want to respond to any concern the reviewer has and to extend our reply if the reviewer would want to elaborate on the core intention of her/his concerns, which we will gladly mitigate.

---

### Official Review · Reviewer_sAtx · 2022-07-13

**Rating:** 6
**Confidence:** 3
**Soundness:** 3 good
**Presentation:** 3 good
**Contribution:** 3 good

**Summary:**

The paper studies the PAC generalization bound and relaxes the uniform bounds assumptions in literature by the on-average bounded loss and on-average bounded gradient norm assumption. Under this relaxed assumption, the paper derives new generalization bound related with the loss-gradient norm. The paper verified its effect empirically.


**Questions:**

The paper should be clearer about the assumption on the data, i.e., how to understand these assumptions and derived bound therein when the real data do not satisfy these assumptions.

**Limitations:**

See the later two parts of the weakness.

**Strengths And Weaknesses:**

Strengths:

The paper relaxed an unrealistic assumption in previous PAC generalization bound. The relaxation is natural and interesting, providing a fine-grained view to improve existing results. The average gradient norm has been studied to reflect the generalization of the algorithm [1], and this PAC bound gives more theoretical evidence on the importance of the average gradient norm.

[1] Understanding generalization error of SGD in nonconvex optimization

It uses clean mathematical language and explains the caveats clearly. Overall the presentation is very good.

Weakness:

The assumptions on the data are strong (x Gaussian or mixture Gaussian, y label balance) in Section 3's entropy bound. It is not clear how critical these assumptions are in terms of deriving a meaningful bound.

 In the generalization bound, it considers the complexity term and ignores the KL term and empirical loss term. Only optimizing the complexity term may lead to some undesired behavior: "one may achieve a fast rate bound by forcing the gradient-norm to vanish rapidly, practical experience shows that vanishing gradients prevent the deep net from fitting the model". This hides the influence of the empirical loss and the KL divergence part.   It would be great if they can be jointly studied, i.e., trade-off the noise variance and lambda.

Though non vacuous, the paper does not demonstrate the power of such generalization bound. It will be much more interesting if we can see how the generalization bound / numerically computed generalization gap changes according to the model architecture, data distribution, or label balance.

---

> ### Author Response · Authors · 2022-08-02
> **Response to reviewer sAtx**
>
> **Mixture of Gaussians** - We think the assumptions on the data are not as strong: We assume the data distribution follows a mixture of Gaussians conditioned on the label, i.e., each subset of the data associated with a label is generated from a different mixture model. In addition, please note that the number of components in the mixture model is not limited. Such a Gaussian mixture model can approximate smooth densities [1, 2, 3]. Hence, we don’t consider this data assumption to be strict. We agree with the reviewer that it is an important discussion, and we will add it to the camera ready.
>
> [1] Statistical Analysis of Finite Mixture Distributions, D. M. Titterington, A. F. M. Smith, and H. E. Makov, 1985
>
> [2] Multivariate Density Estimation: Theory, Practice, and Visualization, David W. Scott, 1992
>
> [3] Deep Learning, Ian Goodfellow and Yoshua Bengio, and Aaron Courville, 2016
>
> **Per-label loss balance** - The per-label loss balance originates from bounding the term $\text{Ent}_{\mathcal{D}_y}[\mathbb{E}\_{x\sim \mathcal{N}_y}[f_w]]$ in Eq. 17. If the loss is per-label balanced, the entropy equals zero. In fact, the per-label balance can be relaxed by assuming that the loss is per-label bounded within an amplitude (See 3.16 in [4]). Please also note that we assume the per-label balance assumption is based on the prior distribution of the parameters. We decided not to present this relaxation in the paper as it complicates the theorems, and as we discuss next, the per-label balance holds in practice.
>
> Please also note that we empirically verify this assumption in Sec. 4 (L233-237). We conducted the following experiment: given a classifier and a dataset, we evaluated the loss value for each label. Using ten different architectures and two datasets, the mean value of the loss of each label was 4.605, and the standard deviation was 0.022. This indicates that the per-label balance assumption holds, as the loss values do not change much across labels.
>
> [4] Probability in High Dimension, Ramon van Handel, 2016
>
> **Complexity term vs. other terms** - We agree with the reviewer that only optimizing the complexity term may lead to undesired behavior. However, we do not perform such optimization since we bound the complexity term using only the prior distribution of the parameters. During training, we only optimized the empirical loss. KL optimization led to worse training performance than without it. We will highlight this point in the camera-ready version.
>
> **More models** - To further demonstrate the use of the bound, we performed a new experiment. We study the effect of important components in ResNet using our bound. For this, we train four variations of the ResNet18 model: 1) a standard model (ResNet); 2) a model without skip connections (ResNetNoSkip); 3) a model without batch normalization layers (ResNetNoBN); and 4) a model without both skip connections and batch normalization layers (ResNetNoSkipNoBN). We optimize all models on the CIFAR10 data:
>
> |Model|Test loss|Train loss|Bound on $C(\sqrt{m}, p)$|
> |--|:--:|:--:|:--:|
> |ResNet|0.722$\pm$0.01|0.541$\pm$0.06|0.185$\pm$0.006|
> |ResNetNoSkip|0.631$\pm$0.02|0.478$\pm$0.05|0.179$\pm$0.009|
> |ResNetNoBN|0.603$\pm$0.05|0.564$\pm$0.08|0.172$\pm$0.007|
> |ResNetNoSkipNoBN|2.302$\pm$0.01|2.31$\pm$0.02|0.01$\pm$0.0004|
>
> We observe that ResNet and ResNetNoSkip achieve comparable performance in all metrics. Additionally, removing the batch normalization layers and including the skip connections achieves comparable performance to ResNet and ResNetNoSkip. Similar to Zhang et al. [5], this finding suggests that even without batch normalization, models can converge using precise initialization. Interestingly, by removing batch normalization and skip connection layers, the model gets to a rate of \lambda=m and achieves a good generalization bound. However, this comes at the expense of poor model fitting to the train set due to gradient vanishing. These results are consistent with prior findings in which batch normalization improves optimization [6]. To conclude, we were able to obtain a tight upper bound of the generalization gap with our proposed bound. However, it is important to note that when using any generalization bound, one should care about the training loss as well as the complexity term.
>
> [5] Fixup Initialization: Residual Learning Without Normalization, Hongyi Zhang and Yann N. Dauphin and Tengyu Ma, 2019
>
> [6] How does batch normalization help optimization?, Santurkar, Shibani and Tsipras, Dimitris and Ilyas, Andrew and Madry, Aleksander, 2018
>
> **Comments** - Great suggestion. We will add the discussion above to the camera-ready.

---

> > ### Comment · Reviewer_sAtx · 2022-08-08
> > **keep the score**
> >
> > The response is not very persuasive, and I will keep the score unchanged.

---

> > > ### Author Response · Authors · 2022-08-08
> > > **Thank you**
> > >
> > > Thanks a lot for reading our rebuttal. If the reviewer would like to elaborate, we are happy to address any remaining concerns.

---

### Author Response · Authors · 2022-08-02
**Thank you**

We thank all reviewers for their time and insightful feedback. We are happy the reviewers agree that our paper is well-written and find it 'interesting', 'meaningful', and 'novel'.

---

### Meta-Review · Area_Chair_G85Q · 2022-08-23

**Recommendation:** Accept
**Confidence:** Certain

**Metareview:**

This paper presents a generalization bound that strengthens influential earlier bounds in a clear and meaningful way.  They provide support experiments.  The paper is well written.


**Award:**

No

---

### Decision · Program_Chairs · 2022-09-14

Accept